# The Chemokine Receptor CCR1 Mediates Microglia Stimulated Glioma Invasion

**DOI:** 10.3390/ijms24065136

**Published:** 2023-03-07

**Authors:** Nazende Zeren, Zobia Afzal, Sara Morgan, Gregory Marshall, Maithrayee Uppiliappan, James Merritt, Salvatore J. Coniglio

**Affiliations:** 1School of Integrative Science and Technology, Kean University, Union, NJ 07083, USA; 2Department of Biological Sciences, Kean University, Union, NJ 07083, USA

**Keywords:** glioma, microglia, macrophage, invasion, CCR1, CCL3, CSF1R

## Abstract

Glioblastoma multiforme (GBM) is the most aggressive form of adult brain tumor which is highly resistant to conventional treatment and therapy. Glioma cells are highly motile resulting in infiltrative tumors with poorly defined borders. Another hallmark of GBM is a high degree of tumor macrophage/microglia infiltration. The level of these tumor-associated macrophages/microglia (TAMs) correlates with higher malignancy and poorer prognosis. We previously demonstrated that inhibition of TAM infiltration into glioma tumors with the CSF-1R antagonist pexidartinib (PLX3397) can inhibit glioma cell invasion in-vitro and in-vivo. In this study, we demonstrate an important role for the chemokine receptor CCR1 in mediating microglia/TAM stimulated glioma invasion. Using two structurally distinct CCR1 antagonists, including a novel inhibitor “MG-1-5”, we were able to block microglial activated GL261 glioma cell invasion in a dose dependent manner. Interestingly, treatment of a murine microglia cell line with glioma conditioned media resulted in a strong induction of CCR1 gene and protein expression. This induction was attenuated by inhibition of CSF-1R. In addition, glioma conditioned media treatment of microglia resulted in a rapid upregulation of gene expression of several CCR1 ligands including CCL3, CCL5, CCL6 and CCL9. These data support the existence of tumor stimulated autocrine loop within TAMs which ultimately mediates tumor cell invasion.

## 1. Introduction

Glioblastoma multiforme (GBM) is an almost invariably fatal cancer with a dismal prognosis of about 12 months from time of diagnosis [1]. One of the hallmarks of GBM is its propensity to locally invade healthy brain rendering complete surgical resection nearly impossible. New therapeutic options are urgently needed and small molecules or biologics which prevent the invasive spread of GBM would represent a major advance. 

Another aspect of GBM is the striking degree of tumor associated macrophage/microglia (hereafter referred to as TAMs) infiltration. In many patients, TAMs can comprise 33% of the total cellular tumor mass [2]. The role of these TAMs were unclear until relatively recently when it was demonstrated by several laboratories, including ours, that they actively promote glioma invasion [3,4,5,6,7]. These studies are consistent with a central role of tumor-infiltrating macrophages in promoting invasion and metastasis which have been experimentally confirmed in most solid tumor animal model [8,9,10,11]. A variety of growth factor ligands are upregulated in the glioma microenvironment and facilitate cross-talk between glioma cells and TAMs. Colony Stimulating Factor-1 (CSF-1) has been shown to be especially important in mediating this interaction within glioma tumors and blockade of CSF-1/CSF-1R pathway has been very effective preventing metastatic progression in many different preclinical cancer models [10,12,13,14,15,16,17,18,19]. Using the GL261 model cell line, we showed in-vitro and in-vivo that blockade of TAMs using Pexidartinib/PLX3397, a blood brain barrier penetrant colony stimulating factor 1 receptor (CSF-1R) inhibitor, strongly suppresses TAM mediated glioma invasion [20]. Around this time it was reported by the Joyce laboratory that PLX3397 was able to fully cure mice challenged with a viral-oncogene mediated gliomagenesis [21]. PLX3397 showed a modest effect in Phase II clinical trials and has been FDA approved for treatment of tenosynovial Giant Cell tumor (now marketed under the trade name Turalio) [22,23]. However, PLX3397 depletes the brain of almost all microglia in normal healthy animals [24]. The CSF-1R is present on all myeloid cells and is essential for normal monocyte development and tissue macrophage homeostasis, therefore its systemic delivery is not an optimal approach for treatment of cancer. Discovery of genes and targets more specifically associated with TAMs as opposed to normal cells of the mononuclear phagocytic system would be a more beneficial strategy.

Chemokines are a large family of secreted proteins which bind to G-protein coupled receptors to mediate various biological effects on target cells [25,26]. The chemokine family consists of a large group of over fifty ligands and receptors which are expressed primarily on cells of the immune system [27]. Four subfamilies of chemokine ligands are determined by the spacing of the two cysteines nearest to the N-terminus. One of the principal roles of chemokines is to mediate chemotactic trafficking to different tissues, however, chemokines have been shown to have pleiotropic effects on cells including regulation of cell division, survival, migration, and invasion. The chemokine receptor C-C receptor 1 (CCR1) is expressed on monocytes and is involved in migration into sites of inflammation [26]. The principal ligand for CCR1 is CCL3 (also known as macrophage inflammatory protein -1a /MIP-1a), however, many other chemokines have been shown to bind to and activate CCR1. These include CCL3,4,5,6,7, and 9/10 in mice and CCL3, 4, 5, 7, 8, 13–16, 23 in humans [25]. It is currently unclear whether these different ligands can elicit specific cellular responses via CCR1. 

In addition to its normal physiological role in leukocyte trafficking, the chemokine system has been implicated in metastatic homing of cancer cells to specific secondary organ sites [28,29]. However, chemokines can play multiple roles in cancer progression including modulating proliferation, survival, invasion, and angiogenesis. The chemokine system is likely to play a pivotal role in determining the makeup of the tumor immune microenvironment including the recruitment of TAMs. For example, the CCL2/CCR2 pathway has already been established in glioma-TAM interaction [30,31]

Here we have investigated the role of CCR1 in TAM-stimulated glioma invasion. Using two structurally unrelated CCR1-specific antagonists we show that CCR1 is necessary for TAM activation of glioma invasion in-vitro. Furthermore, we show that mRNA for CCR1 and CCR1 ligands, including CCL3, are strongly upregulated in a microglial cell line treated with glioma conditioned media. Finally, we provide evidence that CCR1 functions downstream of CSF-1R.

## 2. Results

### 2.1. Pharmacological Inhibition of CCR1 Prevents Microglial-Activation of Glioma Invasion

To investigate the potential role of CCR1 in TAM stimulation of glioma invasion, we utilized a modified version of a Matrigel-coated transwell assay in which fluorescently labelled glioma cells are cocultured with a spontaneously immortalized murine microglial cell line (MG). These assays were carried out in the presence of two structurally distinct pharmacological CCR1-specific antagonists. (Figure 1A). The synthesis of CCR1-specific antagonists has previously reported starting from a pyrrolidine amide lead compound [32]. Of the compounds in that report, we decided to more extensively investigate compound #4 which we now refer to as “MG-1-5”. [32]. The MG-1-5 compound was shown to have good in-vitro potency, and good bioavailability when dosed orally in rats. [32] We compared results of the MG-1-5 compound with the commercially available CCR1-specific antagonist BX471 (Berlex). To examine the effect of CCR1 inhibition on TAM stimulation of glioma invasion, we employed the murine glioma (GL261) and microglia in-vitro coculture model which we previously developed (Figure 2A; [33]). This assay displays robust stimulation of GL261 glioma invasion when cocultured with differentially labelled MG mouse microglial cell line. Consistent with previous studies, coculture of GL261 stably expressing m-Cherry with the murine microglia cell line MG stimulates GL261 invasion by approximately 8-fold (Figure 1C). In the presence of 250 nM of either BX471 and MG-1-5 microglia-stimulated glioma GL261 invasion was blocked by approximately 50 percent (Figure 1C). Using MG-1-5, we measured the dose response to determine the IC50 value. There was a clear dose dependent effect of CCR1 inhibition with 1 uM of MG-1-5 enough to completely inhibit MG-stimulated glioma invasion (Figure 1D). The IC50 for inhibition of invasion for MG-1-5 is 389.1 nM. 

### 2.2. Glioma Conditioned Media Induces CCR1 Expression in Microglia

We next explored if glioma tumor cells could influence expression of CCR1 in microglia. MG cells were starved in MSFM media with 0.3% BSA overnight and then treated with GL261 conditioned media (GL261-CM) for 4 and 24 h. Conditioned media was harvested from confluent monolayer of GL261 cells. Total RNA was isolated from untreated and treated MG cells and used to assess CCR1 gene expression using quantitative real time PCR. As shown in Figure 2A, CCR1 mRNA is strongly enhanced in MG cells treated with GL261-CM (Figure 2A). We next wanted to verify if CCR1 was upregulated at the protein level as well. In order to accomplish this, MG cells were allowed to adhere to glass coverslips and then treated as described above with GL261-CM. Using a CCR1 specific antibody, we were able to detect an enhanced fluorescence intensity of CCR1 on MG cells stimulated for 4 and 24 h with GL261-CM (Figure 2B). These data show that CCR1 expression is rapidly upregulated in glioma associated macrophages.

### 2.3. Glioma Conditioned Media Induces CCR1 Ligand Expression in Microglia

Discovery of high CCR1 levels on tumor stimulated microglia is suggestive this pathway is involved in a paracrine interaction between glioma cells and macrophages. We measured the level of MIP-1a/CCL3 (the principal CCR1 ligand) in GL261 cells by qPCR and ELISA. The cT values of CCL3 in GL261 cell derived cDNA were consistently higher than 32 cycles (data not shown). Consistent with this, we were unable to detect any significant levels of CCL3 secreted by GL261 cells (data not shown) Interestingly however, MG cell stimulation with GL261-CM strongly induces gene expression of CCL3 as well as several other chemokine genes which encode for CCR1 ligands CCL5, CCL6 and CCL9 (Figure 3A). In particular, CCL3 is rapidly upregulated at 4 h post stimulation (Figure 3A). The mRNA for the ligands CCL5, CCL6 and CCL9 are also upregulated and steadily increase up to 24 h. We next performed ELISA analysis on supernatants harvested from MG cells and found a substantial increase in CCL3 secretion from 242 pg/mL in the untreated condition to over 1.5 ng/mL at 4 h post GL261 conditioned media stimulation (Figure 3B).

### 2.4. CSF-1R Signaling Partially Controls CCR1 Expression

We have previously shown that the CSF-1/CSF-1R axis is critical for microglia stimulated glioma invasion using both the coculture in-vitro assay and using syngeneic C57BL/6 mouse model [34]. Recently we demonstrated that CSF-1R mediates invasion in part, via upregulation of the gene Amphiregulin/AREG that presumably acts in a paracrine fashion to induce invasion of glioma cells [34]. We wanted to assess if there is any crosstalk between the CSF-1R and CCR1 pathways in glioma-stimulated microglia. The CSF-1R specific antagonist “JnJ”, is able to partially inhibit the induction of CCR1 mRNA by GL261 glioma conditioned media (Figure 4A). Furthermore, treatment of MG cells for 24 h with 10 ng/mL recombinant hCSF-1 is able to induce expression of mRNA (Figure 4B). This data demonstrated that CCR1 is acting downstream of CSF-1R signaling in TAMs. 

## 3. Discussion

It has been shown that tumor associated macrophages and microglia play an essential role in glioma invasion in-vitro and in-vivo. We have thus far shown with the in-vitro coculture model using murine GL261 cells and MG microglial cells that CSF-1/CSF-1R axis is important for mediating invasion and this is partly dependent on the CSF-1R dependent upregulation of the EGFR ligand Amphiregulin in a paracrine manner. Here, we have uncovered another layer to mechanism by which TAMs promote glioma invasion. Exposure of MG cells to glioma conditioned media strongly induces expression of the CCR1 receptor and several of its ligands including CCL3, 5, 6 and 9. Blockade of CCR1 using pharmacological inhibitors is able to strongly abrogate TAM induced GL261 tumor invasion in in-vitro assays. The importance of CCR1 in mediating glioma invasion was further underscored by the fact that GL261 conditioned media strongly upregulates CCR1 levels and several CCR1-specific ligands (most prominently CCL3) in MG cells. Furthermore, we demonstrated that CCR-1 upregulation is partially dependent on CSF-1R signaling. Together, this data supports a model where glioma cells trigger a chemokine autocrine loop within TAMs which is essential for signaling to glioma cells to promote invasion and malignancy (Figure 5). Future studies utilizing specific blocking antibodies and/or CRISPR-mediated knockdown will investigate which of these ligands are strongly upregulated are involved in promoting invasion. 

There are two reports we are aware of that directly attempt to address the role of CCR1 in glioma pathogenesis. A recent study by Zhang et al. hypothesizes that a paracrine loop involving CCL8 expression by TAMs stimulates stemness and invasion by activation of CCR1 and CCR5 on glioma cells [35]. The possibility of CCL8 acting on TAM-expressed CCR1/CCR5 was not addressed. One of the limitations of the present study is that we cannot definitively claim it is CCR1 on TAMs, as opposed to the GL261 glioma cells which is mediating invasion, since we are using pharmacological inhibition to globally inhibit CCR1 activity. We have been unable to consistently observe CCR1 expression in the GL261 cell line whereas the level of CCR1 expression on macrophages and microglia is readily detectable by qPCR and immunofluorescence (data not shown). However, in order to formally rule out involvement of glioma-expressed CCR1 in microglia stimulated invasion, cell-specific knockout approaches (i.e., RNA interference and CRISPR/CAS9) will be employed to deplete CCR1 from either the glioma cells or MG microglia. It remains entirely possible that CCR1 signaling on both TAMs and glioma cells play a role in invasion. It will also be important to confirm the effect of CCR1 inhibition is observed with human GBM lines as well.

An earlier study measured survival of mice orthotopically injected with the GL261 cell line in CCR1 or CCR5 knockout backgrounds [36]. Interestingly they found that CCR1 ablation had no effect on the pattern of immune cell infiltration in the tumor. The authors observed that knockout of CCR1 had no effect on survival of animals challenged with tumor. Unfortunately, the study did not measure the extent of GL261 invasion into surrounding brain in the CCR1 knockout which we would predict (based on the data presented in this study) would be significantly reduced. The observation that CCR1 knockout does not result in glioma tumor growth is not surprising however given that we have never observed any effect of microglia on the proliferation rate of glioma cells in vitro or in-vivo [20]. We also failed to detect any effect of CCR1 inhibition on viability of GL261 cells or microglia cell line MG (data not shown). We therefore hypothesize that CCR1 is not essential for the recruitment but is necessary for reprogramming of TAMs once they accumulate at the tumor. Our observation that GL261 cells do not express CCL3 and that CCR1 is upregulated in MG cells after exposure to GL261-CM support this hypothesis. 

Recently, there have been multiple publications which indicate CCR1 as a mediator of TAM induced metastasis in several cancer models including breast and colorectal cancer [37,38,39,40,41,42]. In particular, a study by Kitamura et al. showed that CCL3 and CCR1 expression by tumor associated macrophages is associated with an enhanced interaction with breast carcinoma cells and metastatic seeding to lung [37]. In this study, a “chemokine cascade” was discovered in which breast carcinoma cells activate CCR2 on metastasis associated macrophages which controls the expression of CCL3. Genetic deletion of CCL3 or CCR1 prevented pulmonary metastasis in the PyMT breast cancer murine model and thus it was concluded that the CCL3/CCR1 axis activated in macrophages is responsible for metastatic seeding of the lung. Intriguingly, CCR1 knockout macrophages displayed less physical interaction with the carcinoma cells, which suggests the existence of a juxtacrine interaction between carcinoma cells and TAMs. There are also several recent studies in the literature which highlight the role of CCR1 in mediating TAM recruitment and colorectal carcinoma (CRC) metastasis to the lung and liver [40,42,43,44,45,46,47,48]. In the APC/SMAD4 mutant mouse model of CRC, the expression of CCL9 was found to be upregulated in the tumor epithelium and knockout of CCR1 prevented immature myeloid cell (also known as myeloid derived suppressor cells or MDSCs) accumulation at the tumor [47]. This was also validated in a human CRC line which was shown to express another CCR1 ligand, CCL15 [49]. The myeloid cells recruited to the tumor secrete matrix metalloproteinases which facilitate invasion and metastasis to the liver. A pharmacological inhibitor of CCR1 was able to slow metastatic progression and prolong survival of animals challenged with CRC [49]. These findings were consistent with human CRC patient samples where CCL15 expression correlated with the number of CCR1+ MDSCs associated with the tumor [41]. The precise mechanism of CCR1 signaling in tumor associated macrophages in stimulating glioma cell invasion remains to be determined and will be the subject of future studies.

## 4. Materials and Methods

### 4.1. Cell Culture and Reagents

The murine glioblastoma cell line GL261 was obtained from the National Cancer Institute (NCI; Frederick, MD, USA). The murine microglia cell line “MG” was derived from a spontaneously immortalized murine microglia cell culture originally isolated from C57Bl/6J mice as previously described [50]. These cells were originated from primary microglia cultures were generated from high-density mixed cell-type cultures of neonatal neocortex by differential adhesion methods producing highly purified (>99%) microglial populations, as assessed by cell-type specific markers including F4/80, Iba1, CD11b and CSF-1R To further maximize and ensure purity for experiments, the isolated cells were subcultured an additional three times with stringent selective adhesion on nontissue culture-treated “suspension cell” plates (Sarstedt, Newton, MA, USA) to further limit nonmicroglial cells. All cultures were maintained in alpha MEM (Corning Catalog# 45000-300, Glendale, AZ, USA) supplemented with 10% fetal bovine serum (FBS) (Gibco Catalog# 26140079, Amarillo, TX, USA). Microglia were supplemented with 10 ng/mL recombinant mouse granulocyte macrophage-colony stimulating factor (GM-CSF; R&D Systems Catalog #415-ML-10, Minneapolis, MN, USA). All cells were cultured in a humidified incubator containing 5% CO2 at 37 degrees. GL261 cells, a murine glioblastoma cell line which displays invasive properties in-vivo, were cultured in RPMI (Gibco Catalog# 11875093) supplemented with 10% FBS. For inhibition of CCR-1, we used two compounds shown to have specific antagonism to CCR-1 with little to no inhibition of other structurally related receptors (such as CCR5), MG-1-5 and BX471. We prepared MG-1-5 by following the original published synthetic route for compound ‘4’ [32]. The MG-1-5 compound is the most potent of a series of CCR1 antagonists that were published. MG1-5 is a known CCR1 antagonist with an IC50 of 137 nM for CCR1 binding versus CCL3. The BX471 (ZK-811752) compound was purchased from Berlex (Berlex, Montville, NJ, USA). All compounds were dissolved in DMSO at a stock concentration of 10 mM. The CSF-1R inhibitor 4-Cyano-1H-pyrrole-2-carboxylic acid [4-(4-methyl-piperazin-1-yl)-2-(4-127methyl-piperidin-1-yl)-phenyl]-amide (referred to in text as “JnJ”) was obtained from Johnson and Johnson Pharmaceutical Research and Development [51], [52]. Recombinant CSF-1 was purchased from Peprotech (Rocky Point, NJ, USA).

### 4.2. Glioma Invasion Assays

Cell invasion assays were performed as previously described [20,33,34]. Briefly, GL261 cells stably expressing m-Cherry fluorescent protein were cultured alone or with MG microglial cells stably expressing GFP on Matrigel-coated invasion chambers (Thermo Fisher Scientific #354480, Waltham, MA, USA). For most assays, to maintain constant cell numbers, cells were plated at a density per invasion chamber of 150,000 GL261-mCherry expressing cells and 50,000 MG-GFP expressing cells in M-SFM with 0.3% Bovine Serum Albumin (BSA; Sigma Aldrich Catalog #A9647, St. Louis, MO, USA). Imaging of the cells on the bottom of the filter was performed using the EVOS AMG FL microscope. The extent of invasion was quantified by counting the number of fluorescent glioma cells that were on the underside of the filter in at least seven 20X fields. 

### 4.3. Quantitative RT-PCR

Total RNA was isolated from cells in culture using according to manufacturer protocol of Mini RNeasy Universal Kit (Qiagen Catalog# 73404, Hilden, Germany). Briefly, unstimulated MG cells or MG cells stimulated with GL261 conditioned media for 4 or 24 h were lysed with 700 ul Qiazol reagent and subject to chloroform extraction and RNA was purified using RNeasy mini columns. Total RNA was quantified by spectrophotometry and 200 ng was used as a template for cDNA synthesis. Reactions were carried out using the First Strand Synthesis Superscript II Reverse Transcriptase kit (Thermo Fisher Scientific Catalog #18064014) with oligo dT primers (Eton Biosciences, San Diego, CA, USA). The cDNA was subject to quantitative real time PCR using the SYBR dye chemistry method on a Biorad CFX96 Thermocycler with specific primer sets (shown in Table 1). Primers were obtained from Eton Biosciences (Union, NJ, USA).

### 4.4. Immunofluorescence

Cells were allowed to adhere to glass coverslips and then starved in MSFM media overnight. Glioma conditioned media harvested from confluent dishes of the GL261 cell line were used to treat MG cells for 4 and 24 h after which cells on coverslips were fixed in 4% paraformaldehyde in Phosphate Buffered Saline (PBS). Coverslips were then washed several times with PBS 0.1% triton (PBX/Tx). Fixed cells were then blocked using 1% BSA PBS/Tx (IF blocking buffer) for at least 45 min followed by immunofluorescence staining with an anti CCR1 (Abcam Catalog #19013, Cambridge, UK) antibody at a concentration of 1 ug/mL in IF blocking buffer overnight at 4 degrees. The following day, the coverslips were washed 3X with PBS/Tx and stained with a secondary antibody Alexa-Fluor 680 conjugated anti-goat at a dilution of 1:500 in blocking buffer for 30 min at room temperature and then mounted on sides.

### 4.5. ELISA

Supernatants were harvested from either untreated control MG cells or MG cells treated with GL261 conditioned media for 4 h. MG supernatants were centrifuged at 1000 rpm for 5 min in order to remove any suspension cells and analyzed for CCL3 expression using the murine CCL-3 ELISA according to manufacturer instructions (R&D Systems; Duoset Catalog#DY450).

## 5. Conclusions

Here we show that CCR1 is upregulated in glioma associated macrophages and therefore CCR1 antagonists might preferentially interfere with tumor associated macrophages over normal tissue macrophages. Here we extend the findings that CCR1 is involved in cancer invasion and metastasis to glioma, and propose that novel CCR1 antagonists could be very useful for anti-metastatic therapy.

## Figures and Tables

**Figure 1 ijms-24-05136-f001:**
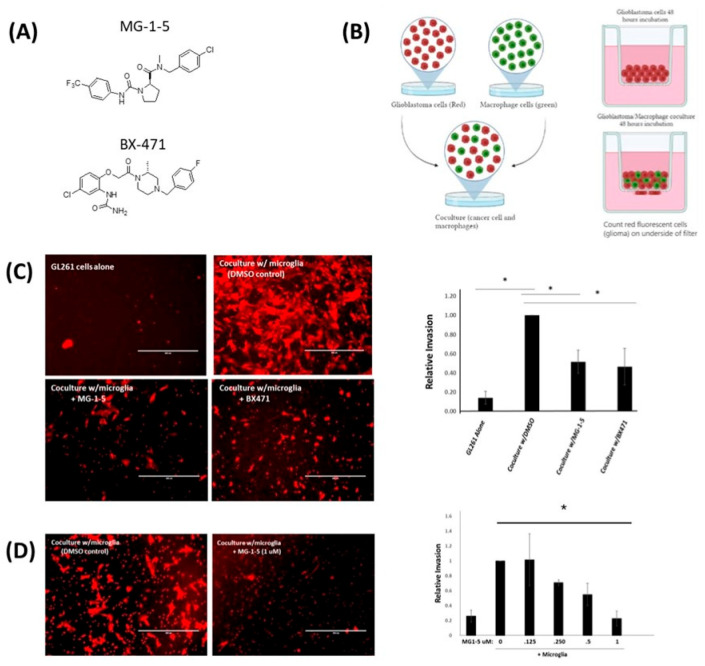
Blockade of CCR1 inhibits microglia stimulated glioma cell invasion. (**A**). Compounds and structures used in the experiment. (**B**) Experimental design of Coculture assay utilized to measure microglia stimulated glioma invasion. GL261-mCherry and MG-GFP (cells not shown) were cocultured on invasion chambers for 48 h in the absence and presence of CCR1 antagonists (**C**). Images and quantitation of invasion assay. * Indicates *p* value of ≤0.05. Data are mean −/+ SEM (n = 3) (**D**) Images of GL261 m-Cherry invasion in DMSO (control) and 1 uM MG-1-5 and quantitation of dose dependent effect of MG-1-5 on glioma invasion.

**Figure 2 ijms-24-05136-f002:**
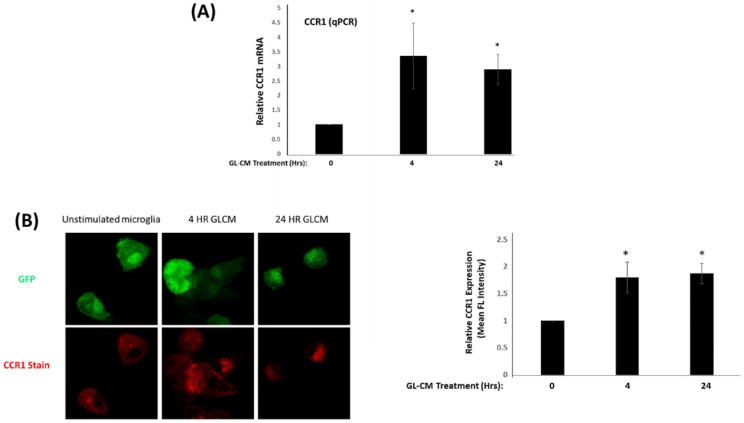
Glioma conditioned media stimulates expression of CCR1 on microglia (**A**) Quantitative PCR of MG cells stimulated with GL261 conditioned media for 4 and 24 h. * *p* value < 0.05. (**B**) Images and quantitation of immunofluorescence of CCR1 in MG cells stimulated with GL261 conditioned media for 4 and 24 h. * *p* value < 0.05. Data are means −/+ SEM (n = 3).

**Figure 3 ijms-24-05136-f003:**
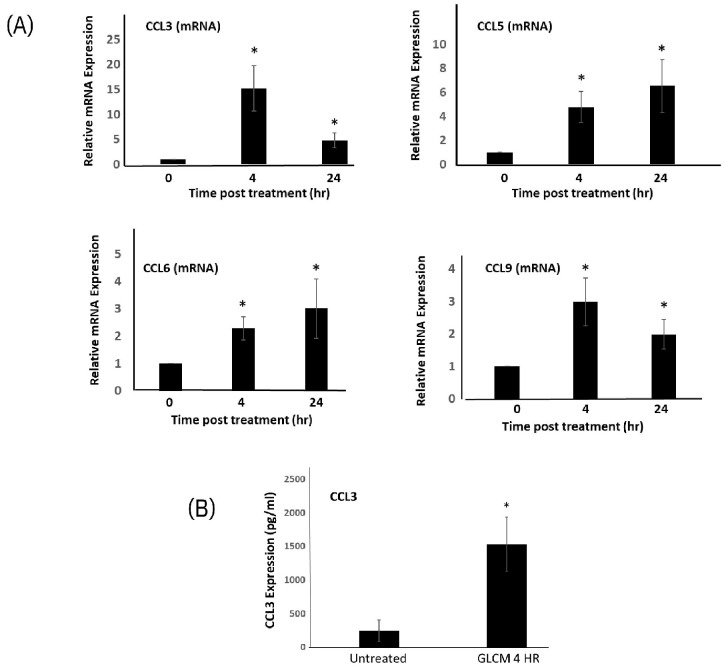
Glioma conditioned media stimulates expression of CCR1 ligands in microglia. (**A**) quantitative PCR was performed on RNA extracted from MG cells treated with GL261 conditioned media for 4- and 24-h using primers specific for CCL3, CCL5, CCL6 and CCL9. * *p* value < 0.05. (**B**) Secretion of CCL3 was evaluated using ELISA on MG cell supernatants treated with GL261 conditioned media for 4 h, *p* value < 0.05. Data are mean −/+ SEM (n = 3).

**Figure 4 ijms-24-05136-f004:**
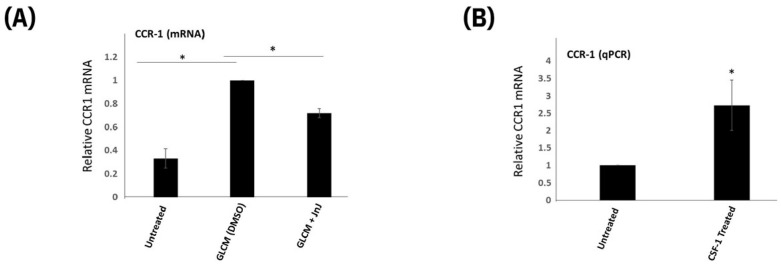
CCR1 expression is downstream of CSF1R. (**A**) quantitative PCR was performed on MG cells treated with GL261 conditioned media for 4 HRs in the absence and presence of the CSF-1R inhibitor JnJ using primers specific for CCR1. (**B**) MG cells were treated for 24 h with recombinant hCSF1 (10 ng/mL) and level of CCR1 mRNA was quantitated using qPCR. * *p* value < 0.05. Data are mean −/+ SEM (n = 3).

**Figure 5 ijms-24-05136-f005:**
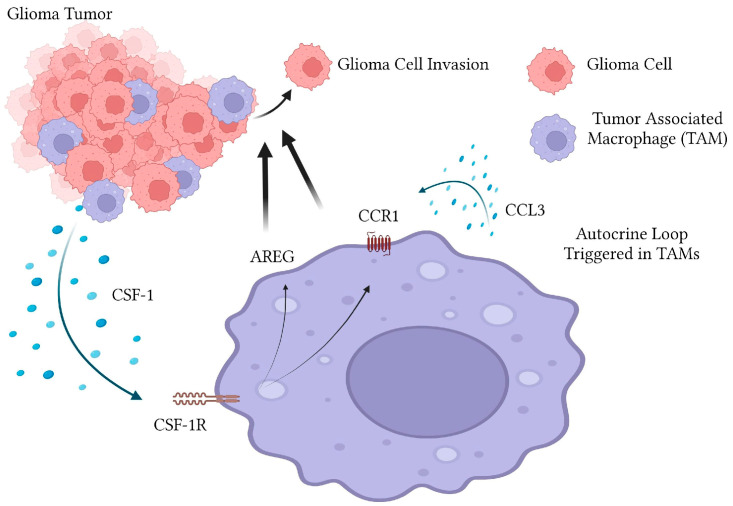
Model of microglia-stimulation of glioma invasion. Glioma cells secrete Colony Stimulating Factor-1 (CSF-1) which activates the CSF-1R and triggers the pro-invasion phenotype of tumor associated microglia/macrophages. This involves upregulation of C-C Chemokine Receptor Type 1 (CCR1) receptor and the EGFR ligand Amphiregulin (AREG). Glioma cells also stimulate upregulation of several CCR1 ligands, including Chemokine (C-C motif) Ligand 3 (CCL3). Blockade of this autocrine loop interferes with small molecule inhibitors of CSF-1R or CCR1 prevents tumor associated macrophage/microglia stimulated glioma invasion.

**Table 1 ijms-24-05136-t001:** Primers used in Quantitative RT PCR assays.

Primer	Sequence
CCR1 Forward	5′-AAGAGCCTGAAGCAGTGGAAG-3′
CCR1 Reverse	5′-GCAGCCATTTTGCCAGTG-3′
CCL3 Forward	5′ ACCACTGCCCTTGCTGTTC-3′
CCL3 Reverse	5′-TCTGCCGGTTTCTCTTAGTCAG-3′
CCL5 Forward	5′-GCTGCCCTCACCATCATCC-3′
CCL5 Reverse	5′-GTATTCTTGAACCCACTTCTTCTCTG-3′
CCL6 Forward	5′-GTGGCTGTCCTTGGGTCC-3′
CCL6 Reverse	5′-AGACCTGGGTTCCCCTCC-3′
CCL9 Forward	5′-CAACAGAGACAAAAGAAGTCCAGAG-3′
CCL9 Reverse	5′-CTTGCTGATAAAGATGATGCCC-3′
GAPDH Forward	5′-CTGGAGAAACCTGCCAAGTA-3′
GAPDH Reverse	5′-TGTTGCTGTAGCCGTATTCA-3′

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
