# Peer review of "The Chemokine Receptor CCR1 Mediates Microglia Stimulated Glioma Invasion"

_ijms, 2023, doi:10.3390/ijms24065136_

Round 1

Reviewer 1 Report

I appreciate the opportunity to review the manuscript “Manuscript ID: ijms-2226757.

“ The Chemokine Receptor CCR1 Mediates Microglia Stimulated Glioma Invasion”.

Authors: Nazende Zeren et al.

Recommendation: Accept with major changes, repeated review.

General comments:

1) It is not clear how murine microglia cell culture (MCC) isolation was performed. Complete the exact description of the MCC isolation.

2) Report the markers for microglia used in the experiment.

3) Explain why murine glioblastoma cell line GL261 was used (and not other lines).

4) Describe all abbreviations in Fig.5 carefully. Complete with an explanation of the mechanism shown in Fig.5

5) Study limitations missing, please add.

6. Language correction is necessary, some sentences are incomprehensible due to syntax and grammar errors

Author Response

We would like to thank the reviewers for their helpful comments. The original comments are underlined followed by our response underneath.

1 It is not clear how murine microglia cell culture (MCC) isolation was performed. Complete the exact description of the MCC isolation.

2 Report the markers for microglia used in the experiment.

To address these points, we have now included in the materials and methods a detailed description of exactly how the MG cell line was derived from murine microglial cultures.  We also listed the markers in the text which were used to verify their status as microglia. (See revised materials and methods)

3) Explain why murine glioblastoma cell line GL261 was used (and not other lines).

The GL261 cell line has several features which are advantageous for studying glioblastoma. This line was derived from and is therefore syngeneic to the C57BL/6 mouse strain. This enables testing in an appropriate in-vivo model (Reference 20). In-vivo, GL261 displays the invasive phenotype which is characteristic of high-grade glioma. In addition to the invasive properties of the GL261, the other major advantage is being able to use these cells in an immunocompetent model.

4) Describe all abbreviations in Fig.5 carefully. Complete with an explanation of the mechanism shown in Fig.5

Abbreviations have been defined and more detail has been provided in the figure legend.

5) Study limitations missing, please add.

We highlighted in the discussion several limitations of the study and future experiments which will address these issues. These include (1) directly measuring the potential of GBM expressed CCR1 in playing a role in TAM stimulated invasion and (2) using human GBM cell lines.

 6 Language correction is necessary, some sentences are incomprehensible due to syntax and grammar errors

We have gone through the manuscript and corrected all grammatical errors.

Reviewer 2 Report

Reviewer comments and suggestions

In this study, the author demonstrated the important role for the chemokine receptor CCR1 in mediating microglia/TAM stimulated glioma invasion. Using a novel CCR1 antagonist, they were able to block microglial activated GL261 glioma cell invasion in a dose-dependent manner. Glioma-conditioned media treatment of microglia resulted in a rapid upregulation of gene expression of several CCR1 ligands including CCL3, CCL5, CCL6 and CCL9. These data support the existence of tumor-stimulated autocrine loop within TAMs which ultimately mediates tumor cell invasion.

Below are the comments for this paper to be incorporated in the revised version of the manuscript. 

  1. Line 68, first time used (MIP-1a) needed full form
  2. Line 70, the reference citation was not correct. 
  3. Can the authors provide details about MG-1-5, such as composition or other? 
  4. It’s not important to write many times same sentence as previous work from our laboratory, better the authors could highlight other research works so that your manuscript could be enriched with other quality
  5. Line 227-228 these studies needed to explain well.
  6. All references need to be modified based on MDPI guidelines.

Author Response

We would like to thank the reviewers for their helpful comments. The original comments are underlined followed by our response underneath.

1 Line 68, first time used (MIP-1a) needed full form

Abbreviation is now defined

2 Line 70, the reference citation was not correct.

We have corrected the citation

3 Can the authors provide details about MG-1-5, such as composition or other?

The MG-1-5 compound is the most potent of a series of CCR1 antagonists that were published in reference 32. MG1-5 is a known CCR1 antagonist with an IC50 of 137 nM for CCR1 binding versus CCL3 (ref. 32). We chose the BX471 compound for comparison as that is the most well established CCR1 specific antagonist in the field. We included more details of this in the materials and methods section.

4 It’s not important to write many times same sentence as previous work from our laboratory, better the authors could highlight other research works so that your manuscript could be enriched with other quality

We have removed all descriptors for “our laboratory”

5 Line 227-228 these studies needed to explain well.

We have greatly expanded the discussion of these studies in this section.

6 All references need to be modified based on MDPI guidelines.

We have corrected the citation style to meet MDPI guidelines.

Round 2

Reviewer 1 Report

The authors responded correctly to the comments. The manuscript can be accepted for publication in present form